

# Incidence and risk factors of systemic lupus erythematosus in patients with primary immune thrombocytopenia: a systematic review and meta-analysis

En-min Zhou, Heping Shen, Di Wang and Weiqun Xu

Department of Hematology-Oncology, Children's Hospital of Zhejiang University School of Medicine, National Clinical Research Center for Child Health, Hangzhou, China

Corresponding author
Weiqun Xu, Vicky__xu@zju.edu.cn

## ABSTRACT

**Background:** Immune disorders and autoantibodies has been noted in both primary immune thrombocytopenia (ITP) and systemic lupus erythematosus (SLE). Whether the two disorders are correlated is unclear. The lack of evidence on the incidence of and risk factors for SLE in primary ITP patients poses a challenge for prediction in clinical practice. Therefore, we conducted this study.
**Methods:** The protocol was registered with PROSPERO (CRD42023403665). Web of Science, Cochrane, PubMed, and EMBASE were searched for articles published from inception to 30 September 2023 on patients who were first diagnosed with primary ITP and subsequently developed into SLE. Furthermore, the risk factors were analyzed. Study quality was estimated using the Newcastle-Ottawa Scale. The statistical process was implemented using the R language.
**Results:** This systematic review included eight articles. The incidence of SLE during the follow-up after ITP diagnosis was 2.7% (95% CI [1.3–4.4%]), with an incidence of 4.6% (95% CI [1.6–8.6%]) in females and 0 (95% CI [0.00–0.4%]) in males. Older age (OR = 6.31; 95% CI [1.11–34.91]), positive antinuclear antibody (ANA) (OR = 6.64; 95% CI [1.40–31.50]), hypocomplementemia (OR = 8.33; 95% CI [1.62–42.91]), chronic ITP (OR = 24.67; 95% CI [3.14–100.00]), organ bleeding (OR = 13.67; 95% CI [2.44–76.69]), and female (OR = 20.50; 95% CI [4.94–84.90]) were risk factors for subsequent SLE in ITP patients.
**Conclusion:** Patients with primary ITP are at higher risk of SLE. Specific follow-up and prevention strategies should be tailored especially for older females with positive ANA, hypocomplementemia, or chronic ITP. In subsequent studies, we need to further investigate the risk factors and try to construct corresponding risk prediction models to develop specific prediction strategies for SLE.

## INTRODUCTION

Immune thrombocytopenia (ITP) is a prevalent bleeding disorder characterized by simple thrombocytopenia and normal or increased bone marrow megakaryocyte counts due to antiplatelet autoantibodies produced by an abnormal T cell response and the proliferation

and differentiation of autoreactive B cells (*Audia et al., 2017*). The annual incidence of ITP is approximately 1.1 to 5.8/100,000 in children and 1.6–3.9/100,000 in adults (*Cooper & Ghanima, 2019*; *Kohli & Chaturvedi, 2019*; *Shaw et al., 2020*).

ITP can be classified into primary and secondary types based on the underlying etiologies. The pathogenesis of primary ITP is unclear, and secondary ITP is associated with autoimmune diseases, immunodeficiencies, infections, and drugs (*Jimura et al., 2018*). Some studies have demonstrated that the incidence of bleeding, thrombosis, infection, cardiovascular disease, and hematological cancer is higher in ITP patients than in the general population (*Hallan et al., 2022*; *Kohli & Chaturvedi, 2019*). Systemic lupus erythematosus (SLE) is a chronic autoimmune disease characterized by autoantibody production and multiorgan involvement, which involves multiple systems and organs, especially the kidneys and central nervous system, resulting in worse prognoses than primary ITP. Several articles have demonstrated that ITP and SLE share a common genetic predisposition (*Fanouriakis, Bertsias & Boumpas, 2020*; *Gkoutsias & Makis, 2022*), and some patients with primary ITP may develop into SLE during follow-up (*Hazzan et al., 2006*; *Song et al., 2022*). However, there are few related studies and the predictive risk factors remain unclear. Therefore, it is pressingly urgent to clarify the clinical features of those patients who develop primary ITP into SLE.

In this study, we analyzed the incidence of and risk factors for SLE in primary ITP patients to provide a scientific basis for clinical treatment and prevention.

## METHODS

### Study registration

This systematic review followed the PRISMA statement. The review program and record were available online through the PROSPERO (CRD42023403665).

### Eligibility criteria
#### Inclusion criteria

1) Articles on patients with primary ITP and excluding secondary ITP;
2) Articles reporting the incidence of SLE or associated risk factors in primary ITP patients;
3) Articles using effective methods for case selection and data collection.

#### Exclusion criteria

1) Non-primary research (*e.g.*, case reports, reviews);
2) Small sample size (<20);
3) Non-English literature;
4) Studies with overlapping or duplicated data.

## Search strategy

Web of Science, Cochrane, PubMed, and EMBASE were searched for articles published from database inception to 7 October 2022 using MeSH terms and free words. To thoroughly include newly published literature, additional searches of these databases were conducted on 30 September 2023. Detailed search strategy is shown in Table S1.

## Study selection and data extraction

The duplicates were excluded using automatic tagging and manual tagging, and then the titles and abstracts of the remaining articles were reviewed for eligible ones for full-text assessment. The full texts of these studies were read for final matches. A standardized table was formulated prior to data extraction. Literature screening and data extraction were carried out independently by two investigators (En-min Zhou and He-ping Shen), then cross-checked, and discussed with a third investigator (Di Wang) in case of disputes.

## Assessment of study quality

The included studies were either cohort studies or case-control studies, so the Newcastle-Ottawa Scale (NOS) was utilized for quality evaluation with eight questions from three domains. The comparability was scored two and the remaining seven questions were scored 1. A total score of 7–9 was considered high quality and a score of 4–6 indicated moderate quality. The NOS-based risk of bias was assessed independently by two investigators, and any disagreements were addressed through negotiation with a third investigator.

## Results

Patients with primary ITP had a higher risk of SLE. Meta-analyses of incidence were performed in cohort studies, and risk factors for SLE in primary ITP patients were reported in a narrative summary.

## Statistical methods

The heterogeneity index ($I^2$) was utilized to reflect the heterogeneity. A random-effects model was adopted when $I^2 > 50\%$ and a fixed-effects model was used when $I^2 < 50\%$. In case of high heterogeneity, sensitivity and subgroup analyses were performed to determine the source of heterogeneity. Funnel plots were applied for visualizing publication bias, and Egger's test was adopted for statistical tests for publication bias. $P < 0.05$ implied statistically significant differences.

# RESULTS

## Study selection

The study selection process is shown in Fig. 1. A total of 3,254 articles were initially screened; after excluding 1,392 duplicates, 1,862 articles were selected for the title and abstract review; 13 studies remained after excluding 1,849 ineligible studies; and eight

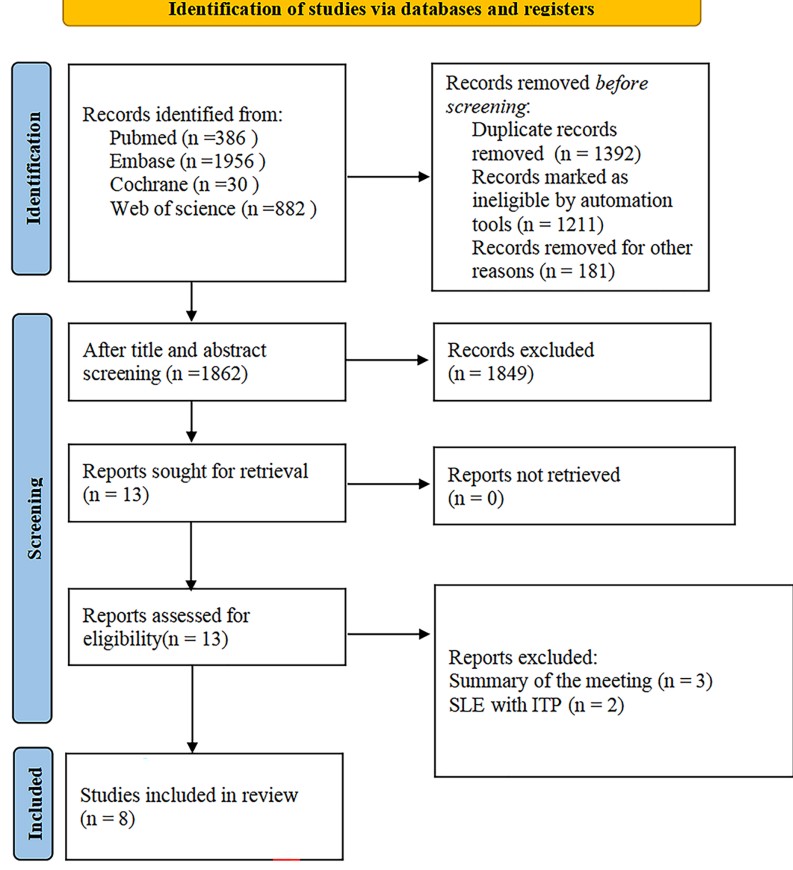

**Figure 1  Study selection process.**         

studies were retained after full-text review (*Adachi et al., 1990*; *Ahn et al., 2022*; *Hazzan et al., 2006*; *Kittivisuit et al., 2021*; *Pamuk et al., 2002*; *Saettini et al., 2021*; *Song et al., 2022*; *Zhu et al., 2020*).

## Study characteristics

The diagnosis of primary ITP was based on the revised guidelines of the ASH working group (*Neunert et al., 2019*), and patients with SLE met the revised 1997 ACR or 2019 EULAR/ACR classification criteria for SLE (*Aringer, 2019*; *Hochberg, 1997*). We included eight articles, consisting of one case-control study and seven cohort studies, mainly from Korea, Turkey, Italy, Japan, Israel, and China. One study in China was multi-centered, and the remaining seven were single-centered. The follow-up period ranged from 6 months to 12 years (Table 1).

## Assessment of study quality

The quality of the eight included articles were assessed using the NOS. Seven of them were single-center studies and therefore had a V3 score of 0 (*Adachi et al., 1990*; *Ahn et al., 2022*; *Hazzan et al., 2006*; *Kittivisuit et al., 2021*; *Pamuk et al., 2002*; *Saettini et al., 2021*; *Song et al., 2022*), and three of them did not analyze the risk factors for subsequent SLE and were

**Table 1 Basic information of included articles.**

| First author | Publication year | Study type | Author country | Sampling time | Patient source | SLE case | Age | Gender | Total cases (ITP) | Follow-up time | Risk factor |
|---|---|---|---|---|---|---|---|---|---|---|---|
| Yuqing Song | 2022 | Case-control study | China | 1990.1–2021.9 | Peking Union Medical College Hospital | 50 | | F:104M:46 | 150 | | Older age hypocomplementemia |
| Soo Min Ahn | 2022 | Cohort study | Korea | 2001.08–2019.11 | Asan Medical Center | 10 | Median: 52 (IQR:34-61) | F:165M:165 | 330 | | Age, organ bleeding, ANA positivity |
| Sirinthip Kittivisuit | 2021 | Cohort study | Thailand | 1976.1–2019.12 | Hematologic Clinic, Department of Pediatrics, Faculty of Medicine, | 14 | 4.2 years | F:256M:217 | 473 | 6.1 ± 6.7 years | Older age chronic ITP |
| Saettini F | 2021 | Cohort study | Italy | 2009.1–2018.12 | San Gerardo Hospital | 1 | Mean: 6.8 ± 4.8 years | F:165M:165 | 330 | 1.1 years | |
| Fangxiao Zhu | 2020 | Cohort study | China | 2000–2013 | National Health Insurance Research Database | 29 | | F:390M:278 | 668 | 80 months | |
| Rawi Hazzan | 2006 | Cohort study | Israel | 1963–2000 | Hematology Oncology Division of the Schneider Children's Medical Center | 8 | 6.7 ± 4.5 years | F:121M:101 | 222 | 4.2 years | Older age femalechronic ITP high ANA titers |
| G.E. Pamuk | 2002 | Cohort study | Turkey | 1984–2000 | Cerrahpaş, a Medical Faculty, Department of Internal Medicine, Division of Hematology | 6 | Median: 34 years (range: 14–78) | F:229M:92 | 321 | 6 months | |
| Masanori ADACHI | 1990 | Cohort study | Japan | 1965–1983 | Research Center of Comprehensive Medicine | 9 | Mean 36 years | F:30M:9 | 39 | 5 years | |

**Note:**

Ahn et al. (2022), Pamuk et al. (2002), Saettini et al. (2021), Hazzan et al. (2006), Adachi et al. (1990), Song et al. (2022), Kittivisuit et al. (2021), Zhu et al. (2020).

**Table 2  Risk of bias in studies.**

| No. | Author | Year | V1 | V2 | V3 | V4 | V5 | V6 | V7 | V8 |
|---|---|---|---|---|---|---|---|---|---|---|
| 1 | Soo Min Ahn | 2022 | 1 | 1 | 0 | 1 | 2 | 1 | 1 | 1 |
| 2 | G.E. Pamuk | 2002 | 1 | 1 | 0 | 1 | 0 | 1 | 1 | 1 |
| 3 | Saettini F | 2021 | 1 | 1 | 0 | 1 | 0 | 1 | 1 | 1 |
| 4 | Rawi Hazzan | 2006 | 1 | 1 | 0 | 1 | 2 | 1 | 1 | 1 |
| 5 | Masanori ADACHI | 1990 | 1 | 1 | 0 | 1 | 0 | 1 | 1 | 1 |
| 6 | Yuqing Song | 2022 | 1 | 1 | 0 | 1 | 2 | 1 | 1 | 1 |
| 7 | Sirinthip Kittivisuit | 2021 | 1 | 1 | 0 | 1 | 2 | 1 | 1 | 1 |
| 8 | Fangxiao Zhu | 2020 | 1 | 1 | 1 | 1 | 2 | 1 | 1 | 1 |

Note:
V1-V8 represent each of the eight scoring items for NOS, which in cohort studies are, in order: adequacy of case definition, representativeness of cases, selection of controls, definition of controls, comparability, ascertainment of exposures, same method of ascertainment for cases and controls, and no-response rate; whereas in case-controls, V1-V8 are, in order: representativeness of exposed cohort, selection of non-exposed cohort, ascertainment of exposure, demonstration that outcome of interest was not present before ascertainment of exposure, comparability, assessment of outcome events, adequacy of follow-up, and completeness of follow-up.
*Ahn et al. (2022), Pamuk et al. (2002), Saettini et al. (2021), Hazzan et al. (2006), Adachi et al. (1990), Song et al. (2022), Kittivisuit et al. (2021), Zhu et al. (2020).*

**Figure 2  Forest plot of the risk of SLE in primary ITP patients.** *Ahn et al. (2022), Pamuk et al. (2002), Saettini et al. (2021), Hazzan et al. (2006), Adachi et al. (1990), Kittivisuit et al. (2021), Zhu et al. (2020).*

scored 0 for comparability (*Adachi et al., 1990*; *Pamuk et al., 2002*; *Saettini et al., 2021*). As a result, five articles were graded as high quality (*Ahn et al., 2022*; *Hazzan et al., 2006*; *Kittivisuit et al., 2021*; *Song et al., 2022*; *Zhu et al., 2020*) and three articles were graded as moderate quality (*Adachi et al., 1990*; *Pamuk et al., 2002*; *Saettini et al., 2021*) (Table 2).

## Meta-analysis

Among the eight included articles, seven explored the incidence of subsequent SLE in primary ITP patients. The meta-analysis of the SLE incidence was performed using a random-effects model, which revealed a pooled incidence of 2.7% (95% CI [1.3–4.4%]; $I^2 = 76\%$) (Fig. 2). Five studies provided gender distribution, with 34 females (34/737) and three males (3/657).

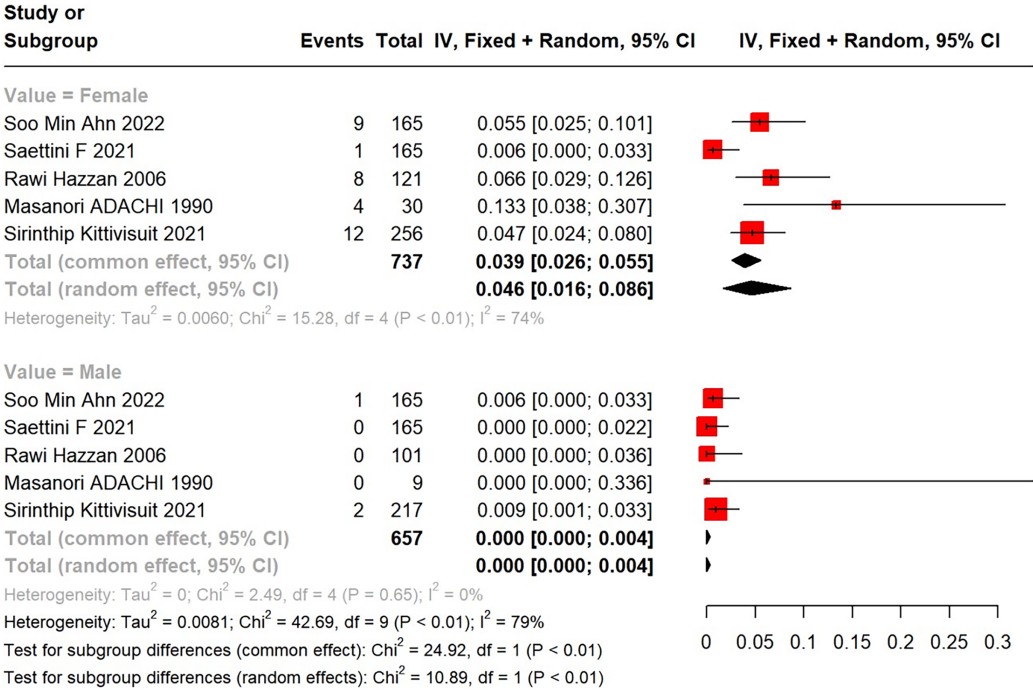

**Figure 3 Forest plot of the risk of SLE in female and male primary ITP patients.** *Ahn et al. (2022),*
*Saettini et al. (2021), Hazzan et al. (2006), Adachi et al. (1990), Kittivisuit et al. (2021).*

(1) Synthesized results

Forest plots of the risk of SLE in primary ITP patients are shown in Fig. 2.

(2) Subgroup analysis

Subgroup analysis by gender unveiled a higher incidence of SLE in female primary ITP
patients than in males. A total of 34 female (34/737) and three male (3/657) ITP patients
developed into SLE. In a meta-analysis using a random-effects model, the incidence of SLE
in female primary ITP patients was 4.6% (95% CI [1.6–8.6%]), with significant
heterogeneity ($I^2$ = 74%), compared with 0 (95% CI [0.00–0.4%]) in males (Fig. 3).

(3) Sensitivity analysis

After excluding each study using the leave-one-out method, meta-analysis results
revealed very good stability and statistical significance (Fig. 4).

## Risk factors

Five studies focused on risk factors, three of which were on children (*Hazzan et al., 2006*;
*Kittivisuit et al., 2021*; *Song et al., 2022*) and two on adults (*Ahn et al., 2022*; *Zhu et al.,*
*2020*), with one study not showing correlations for risk factors using OR and 95% CI
(*Hazzan et al., 2006*) (Table 3). The literature included was limited and few patients in
these articles progressed from primary ITP to SLE. The risk factors varied across studies, so

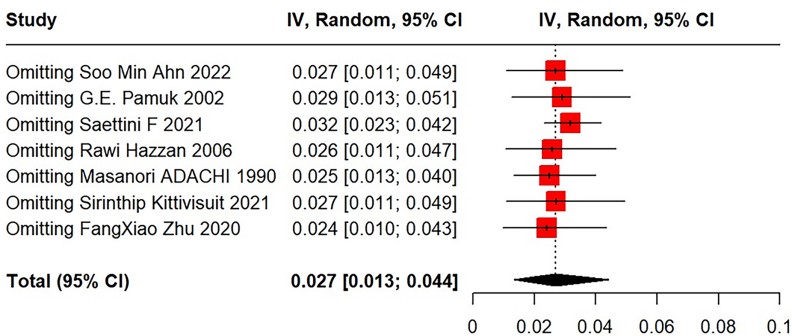

**Figure 4 Sensitivity analysis for SLE in primary ITP patients.** *Ahn et al. (2022)*, *Pamuk et al. (2002)*, *Saettini et al. (2021)*, *Hazzan et al. (2006)*, *Adachi et al. (1990)*, *Kittivisuit et al. (2021)*, *Zhu et al. (2020)*.

**Table 3 Risk factors for SLE in primary ITP patients.**

| Author | Year | Factors | OR | Low | Up |
|---|---|---|---|---|---|
| Soo Min Ahn | 2022 | Age | 6.31 | 1.11 | 34.91 |
| Soo Min Ahn | 2022 | Organ bleeding | 13.67 | 2.44 | 76.69 |
| Soo Min Ahn | 2022 | ANA positivity (≥ 1:160) | 6.64 | 1.40 | 31.50 |
| Yuqing Song | 2022 | Hypocomplementemia | 8.33 | 1.62 | 42.91 |
| Yuqing Song | 2022 | Age | 1.07 | 1.01 | 1.15 |
| Sirinthip Kittivisuit | 2021 | Age | 1.24 | 1.07 | 1.45 |
| Sirinthip Kittivisuit | 2021 | Chronic ITP | 24.67 | 3.14 | 100.00 |
| Fangxiao Zhu | 2020 | Sjogren syndrome | 6.02 | 1.33 | 27.34 |
| Fangxiao Zhu | 2020 | Female | 20.50 | 4.94 | 84.90 |

**Note:**
*Ahn et al. (2022)*, *Song et al. (2022)*, *Kittivisuit et al. (2021)*, *Zhu et al. (2020)*.

multiple regression analyses of risk factors were not performed. Therefore, the risk factors were reported in a review form, which noted that older age (OR = 6.31; 95% CI [1.11–34.91]), positive antinuclear antibody (ANA) (OR = 6.64; 95% CI [1.40–31.50]), hypocomplementemia (OR = 8.33; 95% CI [1.62–42.91]), chronic ITP (OR = 24.67; 95% CI [3.14–100.00]), organ bleeding (OR = 13.67; 95% CI [2.44–76.69]), and female (OR = 20.50; 95% CI [4.94–84.90]) were risk factors for SLE in patients with primary ITP (Table 3).

# DISCUSSION

## Findings

In this study, we discovered that 2.7% (95% CI [1.3–4.4%]; $I^2$ = 76%) of patients with primary ITP developed into SLE. *Zhu et al. (2020)* suggested that ITP patients were 26 times more likely to develop SLE than healthy populations. Extensive research has demonstrated that ITP and SLE have common genetic, pathway, and molecular features (*Fanouriakis, Bertsias & Boumpas, 2020*; *Lee & Bae, 2016*). Genome-wide expression analysis revealed dysregulation of genes involved in major immune response pathways in

ITP, such as T helper cell activation and differentiation, autoantibody response, and complement activation, all in peripheral blood and bone marrow-derived T cells (*Jernås et al., 2013*). Aberrations of immune cells in SLE patients can also be traced back to hematopoietic progenitor stem cells in the bone marrow (*Grigoriou et al., 2020*). Additionally, this meta-analysis concluded that the risk factors for patients with primary ITP developing into SLE during follow-up included female, older age, positive ANA, hypocomplementemia, chronic ITP, and bleeding from internal organs.

## Comparison with previous studies

Our study identified that older age (OR = 6.31; 95% CI [1.11–34.91]) was an important potential risk factor. Older children with primary ITP are more likely to develop into SLE than younger children with primary ITP. The studies by *Kittivisuit et al. (2021)* and *Hazzan et al. (2006)* reported that older age at diagnosis of ITP was a risk factor for the subsequent development of SLE. Song Y found that pediatric patients with primary ITP over 8 years may be at higher risk of developing SLE (*Song et al., 2022*). Considering that age-related alterations in immune function may be involved in the progression of primary ITP into SLE, older children may have an elevated incidence of autoimmune diseases due to the inability to fully tolerate self-antigens (*Kamioka et al., 2020*). Primary ITP patients can develop into SLE at any time during the follow-up, and the longer the follow-up period, the higher the incidence. Sirinthip K revealed that the cumulative risk of developing into SLE at 5 and 10 years after primary ITP diagnosis was 3.8% (95% CI [1.4–6.2]) and 6.5% (95% CI [2.9–9.8]) (*Saettini et al., 2021*). *Song et al. (2022)* reported that the risk of developing SLE increased by 7% every 5 years in primary ITP children. Some studies evidenced that no primary ITP patients developed into SLE during a relatively short follow-up period (mean 2–3.6 years) (*Altintas et al., 2007*; *Moulis et al., 2020*). A study on adults suggested that young adults with primary ITP (<40 years old) were significantly associated with a risk of SLE (*Adachi et al., 1990*). This association may be attributed to the predominance of females of reproductive age (18–49 years old) (*Moulis et al., 2020*), where primary ITP is more prevalent, and SLE is commonly observed among women of reproductive age.

In this study, the subgroup analysis displayed a higher incidence of SLE in female patients with primary ITP (4.6%, 95% CI [1.6–8.6%]; $I^2$ = 74%) than in male patients (0, 95% CI [0.00–0.4%]). *Zhu et al. (2020)* reported that females and Sjogren's syndrome were risk indicators for SLE in primary ITP patients. However, its cause is unknown. Estrogen may be pivotal in increasing susceptibility to autoimmune diseases by stimulating the release of B cells, T cells, macrophages, and cytokines (*e.g.*, IL-1), as well as by reducing apoptosis of autoimmune B-lymphocytes (*Fan et al., 2017*; *Lambert & Gernsheimer, 2017*).

ANA is a crucial diagnostic marker for SLE, Sjogren's syndrome, and other connective tissue diseases (CTDs) and undifferentiated connective tissue diseases (UCTD). CTDs should be differentiated from primary ITP because ANA positivity (1:80) is also observed in some patients with primary ITP (*Altintas et al., 2007*). According to the 2009 ITP standard guidelines, primary ITP patients with ANA positivity may have an increased risk of developing CTDs (*Marmont, 2009*). ANA, as an indicator for SLE diagnosis, has been

focused in previous studies on the risk indicators for primary ITP development into SLE. Nevertheless, its association between ANA and the future development of SLE is controversial in previous studies. A recent study reported a 48-fold higher risk of developing into SLE in ANA-positive primary ITP patients than in ANA-negative ITP patients (*Liu et al., 2021*). SLE development was notably associated with ANA positivity (≥1:160) (OR = 6.638; 95% CI [1.399–31.504]) (*Ahn et al., 2022*). Several studies also exhibited that ANA positivity was a risk index for SLE in primary ITP patients (*Hazzan et al., 2006*; *Pamuk, Ali & Hasni, 2023*). However, *Wandstrat et al. (2006)* found that approximately 25% of healthy individuals were positive for ANA using indirect immunofluorescence assays on HEP-2 cells (*Chen, Lin & Chao, 2021*). ANA is an essential part of normal immune responses, and ANA positivity is also detected upon viral infections, medications (IVIG, *etc*.,), and environmental influences. Many primary ITP children suffer from viral infections or receive immunization 1–3 weeks before disease onset. These children were positive for low-titer ANA and most did not develop into SLE, and ANA turned negative at follow-up (10–28 months later) (*Hazzan et al., 2006*).

Hypocomplementemia (reduction in C3 or C4) is strongly associated with the progression of primary ITP into SLE and should be closely monitored during follow-up. Song Y found that once or more reductions in C3 and C4 levels were a potential risk for SLE in children (OR = 8.33; 95% CI [1.62–42.91]) (*Song et al., 2022*). On the one hand, autoantibodies could activate the complement system by binding to the platelet surface, resulting in hypocomplementemia in primary ITP patients (*Najaoui et al., 2012*). A comparative study manifested that mean C3 and C4 levels were lower in ITP patients than in healthy populations, and about 32% of primary ITP patients might experience at least once complement reduction (*Cheloff, Kuter & Al-Samkari, 2020*). On the other hand, autoantibodies recognize autoantigens to form immune complexes, which in turn activate the canonical pathway of the complement system in SLE patients (*Weinstein, Alexander & Zack, 2021*), with complement depletion, causing a decrease in complement levels. The pathogenesis of primary ITP and SLE may be overlapping, but the association between complement and the risk of primary ITP development into SLE has been less studied. More standardized research in larger sample sizes is needed for further validation.

This study found that the incidence of SLE in children with chronic ITP was 8.3% (OR = 24.67; 95% CI [3.14–100]) (*Kittivisuit et al., 2021*), which was higher than that in overall primary ITP patients (3.02%). *Hazzan et al. (2006)* also reported that only 1 out of 116 children with acute ITP subsequently developed into SLE, while for children with chronic ITP, the figure was increased to 7 out of 106. Moreover, 1–5% of adult primary ITP patients subsequently developed into SLE, but the incidence in chronic ITP patients was about 10% (*Adachi et al., 1990*). Thus, chronic ITP may be a crucial risk for SLE development in primary ITP patients.

Organ bleeding is a serious clinical manifestation of primary ITP and an independent risk for SLE (OR = 13.67; 95% CI [2.44–76.69]) (*Ahn et al., 2022*). In a retrospective study of eight Japanese ITP children with intracranial hemorrhage, three of them subsequently developed into SLE (*Iyori et al., 2000*), but the platelet count level was not correlated with SLE development (*Ahn et al., 2022*). Therefore, further exploration of the risk of bleeding

and mechanisms in SLE will help the proper investigation and prevention of bleeding in SLE.

## STRENGTHS AND LIMITATIONS

This study summarizes the incidence of SLE in primary ITP patients and identifies associated risk factors. The strengths are as follows: first, our summary suggests a higher incidence of SLE in primary ITP patients than in general populations; second, our analysis identifies several potential risk factors for SLE development in primary ITP patients, such as females, older age at onset, chronic ITP, ANA positivity, and hypocomplementemia, which can help physicians identify high-risk individuals and implement appropriate interventions to improve outcomes. Future studies shall explore additional risk factors, develop risk-scoring tools, and create personalized protocols to reduce the incidence of SLE in ITP patients.

However, our study has some limitations. First, the included studies were heterogeneous in study design, study population, and follow-up duration, which might affect the general applicability of our findings. Second, although our search was not restricted to English databases, the included articles were still limited and there might be a selection bias in populations. Third, the included risk factors varied across studies, which might affect the consistency of our study. Fourth, this study only displayed descriptive information without multiple regression analysis of risk factors and failed to quantify the hazard ratios.

## CONCLUSION

This study provides evidence to support the increased incidence of SLE in patients with primary ITP, especially in older, women, ANA-positive, and hypocomplementemia patients. It may be helpful for clinicians to regularly monitor SLE-related indicators in high-risk groups, which can build risk prediction models and develop individualized prevention and management interventions. More standardized research in a larger sample size with a long period of follow-up is needed to further substantiate these results.

### Funding
This work was supported by the Pediatric Leukemia Diagnostic and Therapeutic Technology Research Center of Zhejiang Province (JBZX-201904). The funders had no role in study design, data collection and analysis, decision to publish, or preparation of the manuscript.

### Grant Disclosures
The following grant information was disclosed by the authors:
Pediatric Leukemia Diagnostic and Therapeutic Technology Research Center of Zhejiang Province (JBZX-201904).

### Competing Interests
The authors declare that they have no competing interests.
## Author Contributions

- En-min Zhou conceived and designed the experiments, performed the experiments, analyzed the data, prepared figures and/or tables, authored or reviewed drafts of the article, and approved the final draft.
- Heping Shen conceived and designed the experiments, performed the experiments, analyzed the data, prepared figures and/or tables, and approved the final draft.
- Di Wang performed the experiments, analyzed the data, prepared figures and/or tables, and approved the final draft.
- Weiqun Xu conceived and designed the experiments, authored or reviewed drafts of the article, and approved the final draft.

## Data Availability

The raw measurements are available in the Supplemental File.

## Supplemental Information

Supplemental information for this article can be found online at http://dx.doi.org/10.7717/peerj.17152#supplemental-information.

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
